# Pinpointing Moisture: The Capacitive Detection for Standing Tree Health

**DOI:** 10.3390/s24134040

**Published:** 2024-06-21

**Authors:** Jianan Yao, Zonglin Zhen, Huadong Xu, Liming Zhao, Yuying Duan, Xuhui Guo

**Affiliations:** College of Mechanical and Electrical Engineering, Northeast Forestry University, Harbin 150040, China; yjn@nefu.edu.cn (J.Y.);

**Keywords:** capacitance method, standing tree, moisture content, measuring instrument

## Abstract

Background: the feasibility of the capacitance method for detecting the water content in standing tree trunks was investigated using capacitance-based equipment that was designed for measuring the water content of standing tree trunks. Methods: In laboratory experiments, the best insertion depth of the probe for standing wood was determined by measurement experiments conducted at various depths. The bark was to be peeled when specimens and standing wood were being measured. The actual water content of the test object was obtained by specimens being weighed and the standing wood being weighed after the wood core was extracted. Results: A forecast of the moisture content of standing wood within a range of 0 to 180% was achieved by the measuring instrument. The feasibility of the device for basswood and fir trees is preliminarily studied. When compared to the drying method, the average error of the test results was found to be less than 8%, with basswood at 7.75%, and fir at 7.35%. Conclusions: It was concluded that the measuring instrument has a wide measuring range and is suitable for measuring wood with low moisture content, as well as standing timber with high moisture content. The measuring instrument, being small in size, easy to carry, and capable of switching modes, is considered to have a good application prospect in the field of forest precision monitoring and quality improvement.

## 1. Introduction

Measuring the dielectric constant of standing wood is very important for measuring the water content of standing wood. At present, the dielectric constant is measured using the reflection method or resonance method. These methods usually use time domain reflectometry, network analysis, and other large measuring equipment; they usually use 220 V AC power supply, have poor portability, are expensive, and have a high retrofit cost. It is more suitable for indoor analysis and research, and cannot adapt to the timely measurement needs in the complex environment in the field. It is necessary to find a physical parameter that can include the dielectric constant. Take parallel plate capacitors as an example, as follows:

Assume that the upper and lower plates, as shown in Figure 1, carry a charge of +*q* and −*q*, respectively; the charge is evenly distributed on the opposite side of the two plates, the electric field between plates is uniform, and its electric field intensity is as follows:
(1)$$E=\frac{q}{\varepsilon_{0}\cdot \varepsilon_{r}\cdot S}$$

$\varepsilon_{0}\approx 8.85\times 10^{-12}F/m$, $\varepsilon_{0}=1/4\pi k$ (*k* = 8.9880 × 10^9^, unit: Nm^2^/C^2^).

The plate length is *L*, the width is *D*, and the plate area is
(2)S=D⋅L

The potential difference between two plates is
(3)$$U=Ed=\frac{q\cdot d}{\varepsilon_{0}\cdot \varepsilon_{r}\cdot S}$$

According to capacitance definition,
(4)$$C=\frac{q}{U}$$

The dual plate capacitance is
(5)$$C=\frac{\varepsilon_{0}\cdot \varepsilon_{r}\cdot S}{d}$$

The relative permittivity of air is 1, and water is 81. Bringing these two extreme values into the equation gives a capacitance value between 0.07 and 5.67 pF, thus calculating the capacitance of the probe and the live wood sapwood medium.

The capacitance method is one of the most typical test methods for the detection of water content in standing trees, which works by detecting electrical parameters of the standing tree. Numerous moisture content (MC) measuring instruments developed over the years leverage the high correlation between the MC of the product and its electrical conductivity or dielectric constant [1]. The capacitance method for measuring water content is commonly used in other fields, such as measuring the water content of crops like wheat, corn, rice, and peanuts, as well as crude oil [2,3], and soil [4]. The current capacitance-based method of wood moisture content measurement is widely used in the wood industry. In the field of low moisture content wood, Shaogang Liu [5,6,7] and Guocui [8] have explored a capacitance sensor-based wood moisture content detection system, and the results showed that the capacitance method has a good mathematical model relationship for wood moisture content detection. Additionally, some scholars have ventured into predicting the water content of live standing trees’ trunks using the capacitive method. For instance, Matheny [9] used a capacitive probe calibration sensor combined with the frequency domain reflection method to detect the water content of tree stems. The results show that the species-specific calibration can convert the measured dielectric constant into the volume water content of the species with different densities. The Japanese scholar Tham [10] and others combined the two methods of wood capacitance measurement and near-infrared spectroscopy to predict water content of trees, and to construct different mathematical models of capacitance and spectra, which provide a basis for analytical methods to evaluate other properties of wood and woody materials. All of these studies show that the capacitance method has a good application, not only in the field of low water content wood detection, but it also has great potential in the detection of water content in the trunk of standing trees (high water content).

Studies have shown that the change in the capacitance signal due to the change in wood moisture content is particularly pronounced at low frequency excitation signals [11]. At present, wood moisture content testing equipment is mostly in the laboratory stage, there is still a lack of moisture content measuring instruments applied in the field for measuring live wood trunk moisture content, and it is difficult to measure live wood with high moisture content. Moreover, traditional measuring instruments have limited range and low accuracy. Therefore, it is necessary to study the capacitance-based field instrument for measuring the water content of standing tree trunks. This study takes basswood and fir as the primary research objects, and verifies their performance through laboratory and field experiments, providing technical support for the accurate measurement of water content in a standing tree, and the precise monitoring and quality improvement of forests.

## 2. Materials and Methods

### 2.1. Instrument Design

The equipment development mainly includes a detection probe, temperature acquisition circuit, live tree trunk moisture content signal detection circuit, keypad circuit, LCD display circuit, power supply, voltage regulator circuit, microcontroller minimum system, and software system; the system framework is shown in Figure 2.

The two searching units of the detection probe are equated into a parallel pole plate capacitor [12,13]. The parallel pole plate capacitor consists of a probe, a shell, and a fixed block. The distance between the two probes is 10 mm. The probe is made of a purple copper rod (Carnation Copper Industry Copper Material Manufacturers, Wenzhou, China) with a diameter of 2 mm and a length of 40 mm. There are 10 mm left in the shell for fixing, and the outer shell is made of an industrial high polymer heat shrink tube (Okes Wire and Cable Company, Dongguan, China), which is fixed on the probe after hot processing. The fixed block is machined from a mold and glued to the sensor using a hot melt adhesive. Wood between the pole plates of the capacitor is used as the dielectric. Different dielectric water rates lead to different dielectric constants, which in turn cause changes in capacitance between the parallel pole plates of the capacitor. The capacitance is calculated as
(6)$$C=\frac{\varepsilon \cdot S}{4\pi kd},\;$$
where

*C* is capacitance,

ε is relative permittivity of the medium between the plates,

*S* is area of the parallel plates (unit: m^2^),

*k* is electrostatic constant (*k* = 8.9880 × 10^9^, unit: Nm^2^/C^2^),

*d* is vertical distance between the two plates (unit: m).

The detection probe constituting the capacitor serves as detection input *C_i_*; the water content signal detection circuit uses a 555 multi-harmonic oscillation circuit (Limao Electronic Technology Company, Shenzhen, China), as shown in Figure 3. In the non-stationary mode of operation, the 555 multi-harmonic oscillation circuit outputs a continuous, frequency-specific electromagnetic wave [14]. The frequency of the output electromagnetic wave is determined by *R*_1_, *R*_2_, and *C_i_*. The formula for calculating electromagnetic wave frequency is as follows:(7)$$T\approx 0.7\left(R_{1}+2R_{2}\right)*C_{i},$$
where

*T* is electromagnetic wave frequency (unit: Hz),

*R*_1_ and *R*_2_ are resistance of the circuit (unit: Ω),

*C_i_* is changing capacitance between the parallel poles (unit: F). Duty cycle is determined by *R*_1_ and *R*_2_.

In the testing process, direct measurement of capacitance values is more complicated [15,16,17]. A frequency of the output electromagnetic wave is used to indirectly characterize the change in capacitance due to the measurability of the electromagnetic wave frequency of the 555 multi-harmonic oscillation circuit. To facilitate calculation, the inverse of electromagnetic wave frequency, i.e., the electromagnetic wave period, is used to complete the entire test verification process. Changes in wood moisture content lead to variations in the dielectric constant and, consequently, alterations in the electromagnetic wave of the multi-harmonic oscillation circuit. The moisture content signal from the measured live wood trunk is collected by detection probe and converted into an electromagnetic wave periodic analog signal through a moisture content signal detection circuit. The analog signal is converted into a digital signal through the modular (A/D) converter of a minimum system of a single chip microcomputer. The STM32F103VET6 (Xintai Electronics, Shenzhen, China) is the microcomputer, and a series of functions, such as multi-resonant circuit output signal measurement, temperature signal processing, key selection measurement mode, calibration formula input and modification, zero calibration, and serial communication, are realized in the burning program using the 12-bit AD converter of STM32. A digital signal is put into the moisture content calculation formula to calculate real moisture content of the live wood trunk, and the final result is displayed on a Liquid Crystal Display (LCD) screen.

The uneven surface of the bark and different sizes of tree diameters directly influence the insertion process of two searching units’ detection probes. In order to ensure that the probe can be easily inserted into the live standing wood, and to reduce the interference of the magnetic field on the detection circuit, the material used for the detection probe was selected with a certain toughness and stiffness of purple copper T2. To prevent short circuiting during insertion, an insulation layer cover was used. The temperature detection circuit sensor utilizes a DS18B20 chip, externally powered, with advantages including high system accuracy, strong resolution, and good anti-interference performance [18]. The key part uses three independent keys, RESET, Key0, and Key1, which occupy three IO ports, WK UP, KEY0, and KEY1 of the microcomputer, respectively, whereas RESET (WK UP) controls the reset of the microcomputer, and the other two (KEY0, KEY1) are mode selection keys, which can enter a selection state by long press. Figure 4 shows key schematics of this design. For the modular design of the LCD display, a 2.4 inch TFT LCD display is integrated into the dedicated interface of the microcontroller minimum system board, with required registers configured and initialized.

Firstly, the main program design involves initializing internal resources, such as an analog to digital converter (ADC) conversion module and clock configuration of the microcontroller and peripheral modules, such as a keypad module and LCD display, and the user selects the appropriate mode according to the detected tree species. Each tree species corresponds to a specific fitting equation, and the fitting equation is written into the subroutine corresponding to each keypad. In each processing cycle, the detected signal is substituted into the fitting equation to derive the water content value, which is then displayed on the LCD screen.

Next is the interrupt program design. The moisture content signal is collected by detection probe and processed through detection circuit, resulting in a periodic electromagnetic wave signal. Secondly, frequency is determined by measuring the number of falling edges of the electromagnetic wave within a unit time, followed by the calculation of the electromagnetic wave period. When the falling edge of electromagnetic wave signal is generated, the external interrupt service is triggered, leading to the entry into the external interrupt program. Upon the occurrence of a falling edge in the electromagnetic wave, the variable of “times” is incremented. When the time “t” reaches 1 s, the timer’s interrupt service is triggered, initiating the entry into the timer interrupt program. The value of the variable “times” is then assigned to the variable of ”last times”, the variable of “times” is cleared to zero, and the total number of falling edges of the electromagnetic wave within a 1 s interval is recorded. Exiting the interrupt program, the system re-enters the main program to proceed with the next 1 s timer interrupt process. The value of “last times” can be used to calculate the frequency of the electromagnetic wave, and then calculate the period of the electromagnetic wave.

The experimental data obtained after A/D conversion will have certain errors, and data collected by the microcontroller need to be optimized to reduce errors brought by the experiment and outside world, and to improve detection accuracy. The data collected within 1 s are arranged from smallest to largest by comparing them one by one, and are stored in an array for easy recall, as follows:a[0], a[1], a[2], a[3] …… a[n], (8)
remove the first four numbers and last four numbers of experimental data and rearrange the entire array as follows:a[0], a[1], a[2], a[3] …… a[n − 8], (9)

The error caused by the experimental interference can be eliminated by removing extreme values. In addition, noise inevitably interferes with the circuit during the testing process. In order reduce this unavoidable global error, the method of averaging data is used for filtering processing, and, finally, the data are stored in a defined variable.

### 2.2. Measuring Standing Tree Area

Sapwood and heartwood are two important zones in the wood. Sapwood has a higher water content, and because sapwood is outside the tree, it usually has a higher water content; this is because sapwood is the main channel for trees to transport water and nutrients, so it contains a lot of water. Heartwood is located in the center of the tree, near the inside of the tree. Heartwood usually contains less water than sapwood; this is because heartwood does not contain living cells, and the water and nutrients have been removed or converted into other substances. Due to the low moisture content in the heartwood and the dense wood structure, the moisture content of the heartwood is relatively stable and not susceptible to seasonal and climatic changes. Therefore, the sapwood of trees is chosen as the research object.

### 2.3. Instrument Verification

To verify reliability of the measuring instrument, the instrument was first calibrated in a laboratory, including using the distilled water medium test and the wood medium test, and the experimentally fitted relationship equation of the electromagnetic wave period and water content was imported into the microcontroller of a detector. Then the instrument was field tested in an experimental forestry field in Liangshui, Yichun. The equipment measurement procedure is as follows: 30 basswood trees and 15 fir trees were randomly selected, and the bark was peeled at a height of 1 m from the ground. A small electric hand drill is used to drill two parallel small holes with a diameter of 3 mm and a depth of 4 cm in the vertical direction (1 cm apart). A small hole of the same size is drilled with a growth cone next to the lower hole, and the sapwood sample (wood core) is taken out and weighed with an electronic balance (to record the total weight of the wood core) and bagged for use. The device is inserted into the tree, with a wait time of 30 s to 1 min, until the data no longer fluctuate; reading and recording the input square wave period is obtained by the water content detector.

Figure 5 shows the physical diagram of the standing tree moisture content meter, which is divided into the following four parts: searching units, detection circuit, lithium battery, and minimum system board. Figure 5a shows the front view of the instrument, and Figure 5b displays the field measurement of the standing tree trunk moisture content meter. The measuring instrument is located 1 m above the ground (the area selected by the red line is the measuring instrument).

According to the principle of the measuring instrument, the period of electromagnetic wave output of the device is affected by moisture content of the wood; a distilled water medium test can be used as a pretest to observe the sensitivity of the test device to water and the degree of influence of the insertion depth change on the period signal, as a blank control for subsequent tests. The distilled water experiment is shown in Figure 6, and test results are shown in Figure 6b. Figure 6b shows that the electromagnetic wave period becomes longer as insertion depth increases. Correlation coefficient R^2^ of the fitted curve is as high as 0.99, indicating that the independent variable has a good linear relationship with the dependent variable when testing distilled water media, and the meter has good sensitivity to distilled water. The distilled water medium test can provide a good reference basis for the detection of water content of live tree trunks in the field.

### 2.4. Instrument Calibration

Laboratory wood media tests were conducted using water-soaked basswood and fir disc specimens instead of live timber. First, two holes were drilled in each of the specimens, which were then soaked in water for four weeks. Subsequently, the disc specimens were removed once saturated, wiped dry, and left to stand for 1 h. The searching units of the probe were then inserted into the drilled holes at depths of 30 mm and 40 mm, respectively, to monitor and record the electromagnetic wave cycle in real time. Readings were recorded every 30 s for a total of five times. After data recording, the specimens were placed into an oven for drying at 105 °C, where they were dried for half an hour each time for high moisture content (>30%), and for 1 to 3 h each time for low moisture content (≤30%) [19]. After each drying, the weight of the specimens was measured and left to dry for 1 h in the room (room temperature of 15 to 20 °C), and weighed and recorded (accuracy 0.1 g). According to the same test steps, the output electromagnetic wave period is detected, and the test data are recorded. These steps were repeated until the specimen reached a constant weight. The absolute dry mass was weighed, and the moisture content was calculated for each test.

In the same experimental site in Liangshui [20], basswood and fir standing trees were randomly selected; 30 basswood trees and 15 fir trees were sampled. Two holes were drilled in each test tree, moisture content was checked with a measuring instrument, and test data were recorded; at the same time, the wood core was removed with a growth cone, oven-dried, and the moisture content of the trunk of standing trees was calculated and compared to the data measured by the measuring instrument.

## 3. Results

### 3.1. Wood Media Test

Figure 7 shows the fitted relationship between the moisture content and electromagnetic wave period for basswood and fir, and fitted equations are shown in Table 1; R^2^ of the fitted equation for the basswood disc at 40 mm insertion depth was as high as 0.99, and the model fit was good and reliable (Figure 7a). The fitted curve of the fir disc specimen showed obvious segmentation (Figure 7b), and the moisture content of the cut-off point was around 30% (near the wood fiber saturation point). The fitted equations of fir disc specimens were better fitted at an insertion depth of 30 mm, and R^2^ of both fitted curves was greater than 0.95, while fir discs had low accuracy and larger errors at the insertion depth of 40 mm. Therefore, fitted equations under 40 mm and 30 mm insertion depths were selected for basswood and fir, respectively, and written into the microcontroller to facilitate field tests for verification.

### 3.2. Field Test Validation

The actual water content of trees was compared to the results obtained by the measuring instrument, and comparison results are shown in Figure 8. Due to the complex measurement environment, measurement results of the meter were slightly lower than indoor measurement results. The R^2^ values of fitted curves of basswood were all greater than 0.8, and measurement results were more satisfactory (Figure 8a), while R^2^ of fitted curves of fir were all greater than 0.95 (Figure 8b), and the fitting effect was better than that of basswood, as shown in Table 2. The average error in water content measured by the two methods was calculated and shown to be 7.75% for basswood and 7.35% for fir, and a comparison of the data is shown in Figure 9. The fitted curves of the capacitance method were better than the drying method for both tree species in Figure 9, indicating that the fitted curves of the capacitance method had some credibility, produced less fluctuation in the measured data, and had higher accuracy of detection with less errors. Therefore, the capacitance method holds practical value in predicting the moisture content of live standing tree trunks.

## 4. Discussion

In this study, the moisture content of two species was studied in the laboratory and field, and the function equations of the electromagnetic wave period and moisture content were constructed for the two species of basswood and fir. This moisture content detector is small in volume, and the measurement is accurate and can display the electromagnetic wave reflection time and moisture content in real time. Compared to the relevant studies of SK Korkua (2020) [21] and P Chetpattananondh (2017) [22], we can write calibration equations of multiple tree species into the device in advance, so that different tree species can be easily measured in actual measurements.

## 5. Conclusions

Based on the capacitance method, we designed a live tree trunk water content meter, constructed an equation of the electromagnetic wave period as a function of water content through indoor tests, and wrote it into a microcontroller, and then used it to measure water content of live tree trunks in the field to verify the reliability of the meter. Conclusions show the following:The equipment is optimized from both hardware and software aspects. Due to the good conductivity of purple copper, it is chosen as the probe material, which can better retain the integrity of an electrical signal; the microcontroller judges pulse the trailing edge to determine that the frequency is stable and reliable; and the 555 timing circuit has high sensitivity to capacitance changes. The software structure is optimized in terms of sampling, especially software filtering, which reduces the occurrence of accidental errors and improves accuracy of data.Pretests on distilled water prior to laboratory testing of moisture content of live tree trunks showed that the electromagnetic wave period gradually increased with the increase in insertion depth, i.e., the increase in moisture, and the goodness of fit was 0.99, indicating that the meter has excellent sensitivity to water.The field and indoor tests showed that applicability of the moisture content meter for standing tree trunks was verified by constructing mathematical models for different tree species. The instrument is capable of measuring moisture content of basswood and fir tree trunks in the range of 0 to 180%, with a wide range of measurements. The average error of measurement was 7.75% for basswood and 7.35% for fir, both within 8%. Overall, the design and measurement results of the instrument were satisfactory.

The moisture content measurement instrument designed in this study can achieve more accurate measurements of the moisture content of standing wood trunks with minimal error, applicable to both low and high moisture content live standing wood trunks. An accurate capacitance moisture content detector is designed, which is convenient to carry, small in size, and can be viewed in real time; furthermore, the preset mode can be imported into the device in advance, and different modes can be switched at any time.

We still need to conduct a lot of experiments to make the calibration equation more accurate. In the future, different tree species will be calibrated and input into the device. Secondly, we can also study the water content at different locations and heights of the same tree, which helps us to make more accurate measurements. It is also necessary to add storage capabilities to the device later on, so that the tree can be measured in the form of a record for a long time.

## Figures and Tables

**Figure 1 sensors-24-04040-f001:**
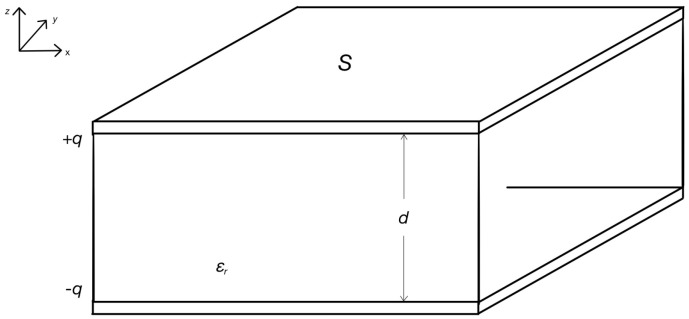
Plane−parallel capacitor.

**Figure 2 sensors-24-04040-f002:**
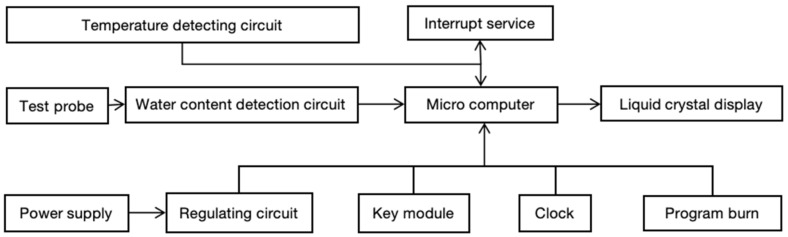
Block diagram of the system of standing tree trunk moisture content measurement.

**Figure 3 sensors-24-04040-f003:**
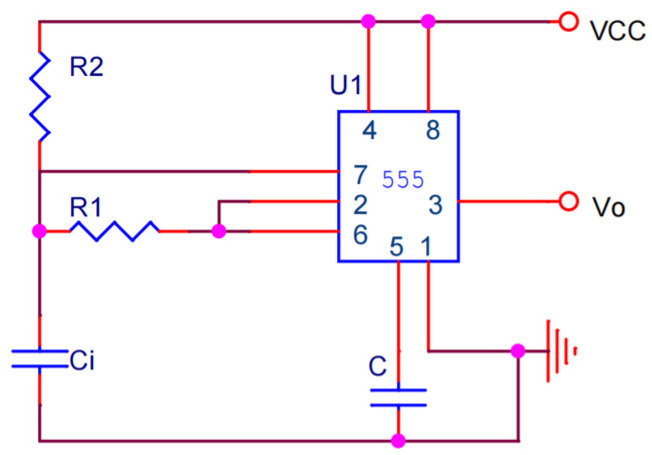
555 multiple resonant swing circuit diagram.

**Figure 4 sensors-24-04040-f004:**
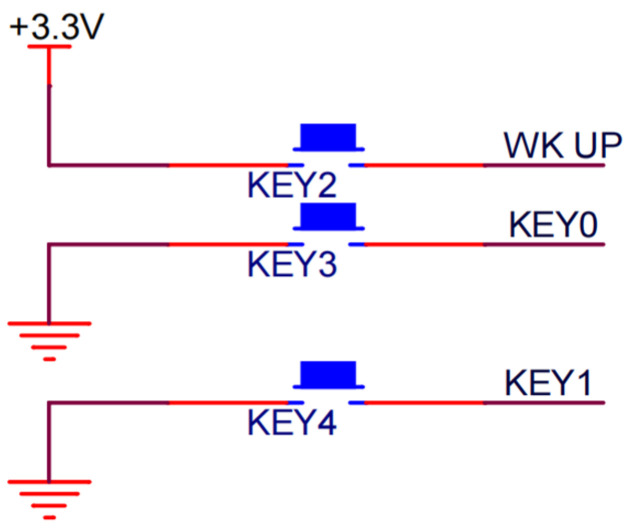
Key schematic diagram.

**Figure 5 sensors-24-04040-f005:**
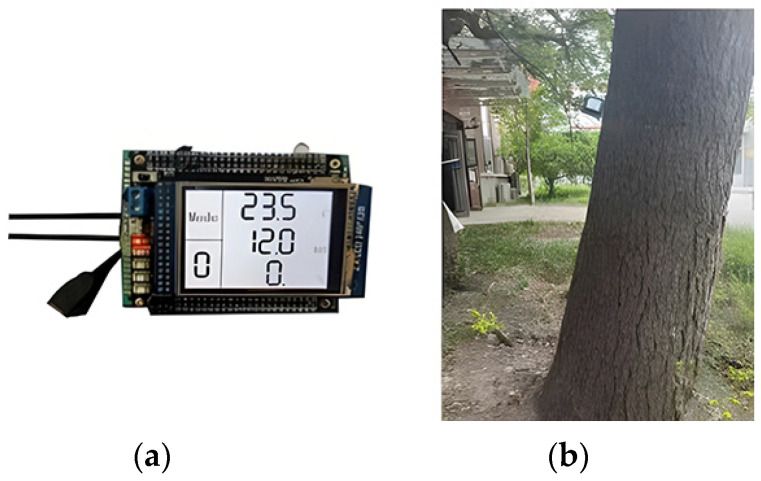
Standing tree trunk moisture meter. (**a**): front view of the instrument. (**b**) field measurement of tree moisture content.

**Figure 6 sensors-24-04040-f006:**
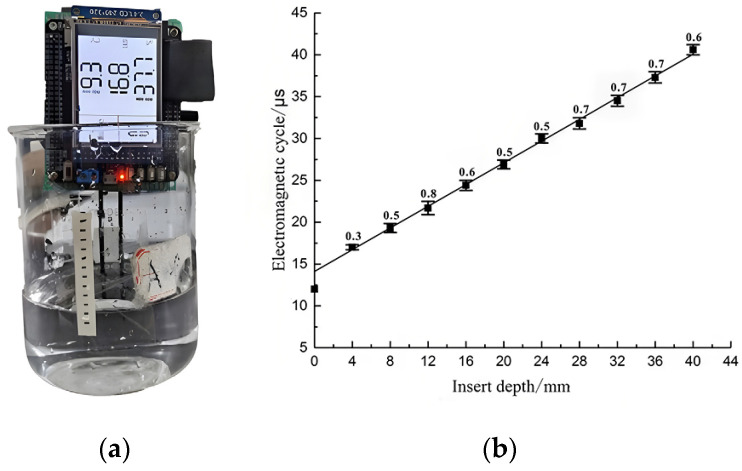
Relation diagram of insertion depth and electromagnetic wave period in water medium test. (**a**) Distilled water medium test; (**b**) fitted curve of the average of six experiments.

**Figure 7 sensors-24-04040-f007:**
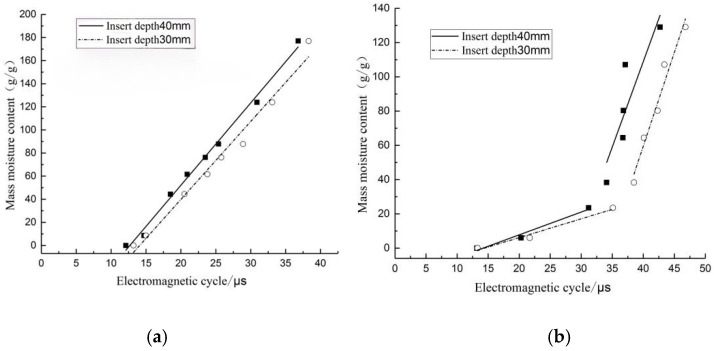
Fitting diagram of electromagnetic wave period and water content of basswood and fir. (**a**) Basswood; (**b**) fir.

**Figure 8 sensors-24-04040-f008:**
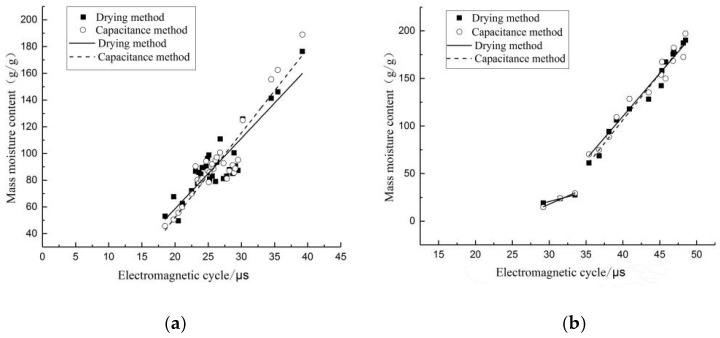
Comparison diagram of water content fitting curve between capacitive method and drying method of basswood and fir. (**a**) Basswood; (**b**) fir.

**Figure 9 sensors-24-04040-f009:**
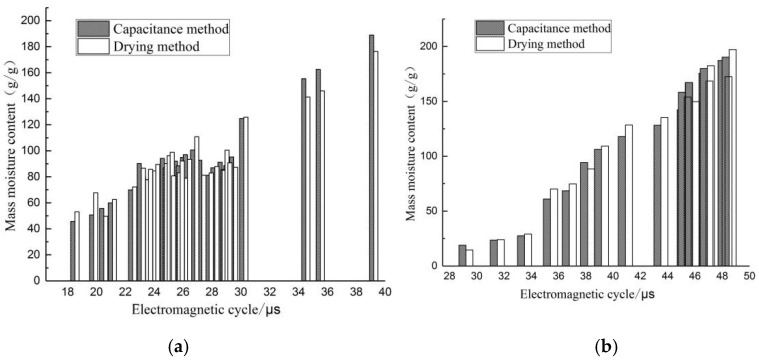
Comparison of basswood and fir measurements with real data. (**a**) Basswood; (**b**) fir.

**Table 1 sensors-24-04040-t001:** Fitting relation between electromagnetic wave period and mass moisture content of basswood and fir.

Tree Species	Insert Depth	Fitting Equation		R^2^	
Basswood	30 mm	*y* = 6.75*x* − 95.10		0.98	
	40 mm	*y* = 7.13*x* − 90.39		0.99	
Fir		Moisture content is less than 30%	Moisture content is more than 30%	*R* _1_ ^2^	*R* _2_ ^2^
	30 mm	*y* = 1.10*x* − 15.85	*y* = 10.92*x* − 377.31	0.98	0.96
	40 mm	*y* = 1.33*x* − 18.87	*y* = 9.99*x* − 290.52	0.98	0.79

Note: *R*_1_^2^ corresponds to the fitted curve with water content less than 30%, and *R*_2_^2^ corresponds to the fitted curve with water content less than 30%.

**Table 2 sensors-24-04040-t002:** Fitting relation between electromagnetic wave period and water content of basswood and fir in different methods.

Tree Species	Insert Depth	Processing Method	Fitting Equation		R^2^	
Bass wood	40 mm	Drying method	*y* = 5.27*x* − 46.90		0.81	
		Capacitance method	*y* = 6.33*x* − 74.50		0.87	
			Moisture content is less than 30%	Moisture content is more than 30%	*R* _1_ ^2^	*R* _2_ ^2^
Fir	30 mm	Drying method	*y* = 3.40*x* − 84.34	*y* = 9.03*x* − 250.87	0.98	0.96
		Capacitance method	*y* = 2.00*x* − 39.29	*y* = 9.73*x* − 282.78	0.99	0.98

Note: *R*_1_^2^ corresponds to the fitted curve with water content less than 30%, and *R*_2_^2^ corresponds to fitted curve with water content less than 30%.

## Data Availability

Data are contained within the article.

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
