# Peer review of "Pinpointing Moisture: The Capacitive Detection for Standing Tree Health"

_sensors, 2024, doi:10.3390/s24134040_

Round 1
Reviewer 1 Report
Comments and Suggestions for Authors
The authors present a measurement device for the determination of the moisture content in wood using a capacitive measurement principle. The work includes a description of the measurement concept based on an oscillator principle and discusses implementation details of the measurement device. In chapter 3 the authors present experimental results obtained from lab tests and field tests.
Overall, the work lacks significant originality with respect to the state of the art. Moisture sensors based on the proposed principle are readily available for a variety of different applications ranging from sensing biological material, wood, concrete etc. Commercially available sensors also show similar accuracy while beeing also tested under harsh environmental conditions (e.g. temperature changes etc.). Therefore the statement on page 2, line 51 with respect to "traditional measuring instruments" is not convincing for me unless the authors provide a profound comparison with state of the art instruments. To my opinion, the applied measurement concept follows the well known oscillator principle, as it is also described in textbooks like "Capacitive Sensors: Design and Applications" by Larry K. Baxter. It remains unclear, while the presented approach would lead to improved accuracy and measurement range.
The contents presented in Section 2 is not about Materials or Methods. It is more a documentation of the device, i.e. the hardware implementation of the device including a loosely coupled description of firmware aspects, hardware components (e.g. the key schematic diagram). The block diagram in Figure 1 shows a general view of the system, without revealing any specifics of the actual measurement probes.
The experimental results are sound and comprise a basic validation of the device and application specific measurements obtained with different wood samples. However, a field deployable measurement device operates under different environemental conditions (e.g. temperature changes). Industrial humidity sensors typically apply complex on-line calibration strategies to improve robustness against these temperature influences. What methods are applied in your device?
The paper needs a careful revision also from the linguistic point of view. E.g. some sentences in the abstract are not complete, e.g. first sentence from line 7 - line 9. The abstract should not include a number labeling of the statements.
Minor editing of English language required
Reviewer 2 Report
Comments and Suggestions for Authors
1. The paper mentions a capacitor with parallel plane surfaces, but it is not described as used dimensions, materials used for plates.
2. The paper mentions a modular ADC converter, but the type of module and its number are not specified. The type of microcontroller used is not mentioned in the work either.
3. I suggest that the schematic diagram be redone with all the components used (555 circuit, ADC module, microcontroller, keys, power, etc.) and use only one figure for the diagram.
4. I suggest that the main program used to operate the scheme should also be included in the paper.

Author Response
Dear reviewer:
Thank you for your decision and constructive comments on my manuscript.We agree with the reviewers’ suggestions and will incorporate the recommended changes into the manuscript.Each comment will be directly addressed regrading the modified manuscript with changes highlighted in yellow.
For your forth suggestion, We put the main program at the end of the article, and if the layout permits, we will put it in our manuscript.
For the other suggestions, we reflected the changes in the manuscript.

Reviewer 3 Report
Comments and Suggestions for Authors
In the summary I consider it necessary to include more information regarding the methodology, for example, to use the meter it is necessary to strip the bark of the tree, how the humidity measurement is carried out, how much the humidity detector penetrates into the wood.
In the methodology, I consider it important to inform how the installation procedure on the tree would be for the device to measure humidity, preferably placing an illustrative photo that allows visualizing the location on the tree (which height will be better) I also consider it interesting to carry out this measurement in different parts of the tree. tree trunk, what are the dimensions of the sensors that will be penetrated into the wood and the effect of electromagnetic emissions that can confer the humidity of the wood, for what time the device should be in the wood, is it necessary to drill the wood before insert the device. It is known that wood in its anatomical formation has two fundamental parts, sapwood and heartwood, in each of these parts the moisture contents are different, thus, I consider it important to define the depth at which the device makes measurements in the wood. To certify this anatomical information of the wood, I consider it important to carry out a characterization of the two species under study.
The introductory paragraph of the results of the work is part of the methodology, since it explains the previous suggestion in a little more detail. You can also place figure 4 in the methodology of the work as a reference.
The simulation carried out in relation to the laboratory tests with wood samples submerged in water for 4 weeks, was carried out based on which standard or justification, do you consider that 4 weeks is sufficient to simulate the real humidity of a standing tree? What was the moisture content of wood samples after that time in water??
I consider that the authors should verify and separate the information that corresponds to the methodology, results and conclusions, given that in the work they appear mixed in items that do not correspond, this can make the reading more extensive and confusing.
Author Response
Dear reviewer:
Thank you for your decision and constructive comments on my manuscript.We agree with the reviewers’ suggestions and will incorporate the recommended changes into the manuscript.Each comment will be directly addressed regrading the modified manuscript with changes highlighted in blue.

Reviewer 4 Report
Comments and Suggestions for Authors
Dear Authors,
please find enclosed all the suggestions and comments on this paper.

Comments on the Quality of English LanguageThe English language should be checked by a native speaker.
Author Response
Dear reviewer,
First of all, we sincerely thank you for your valuable comments and suggestions on our manuscripts. Your guidance is very important to the improvement of our research and papers. All changes made to the text are highlighted in green for easy identification. We have changed the title to ‘Pinpointing Moisture: The Capacitive Detection for Standing tree Health’, the introduction shall be extended and modified, and the contents of Materials and Methods chapter shall be adjusted and modified. The fourth chapter Discussion is added. The content of Chapter Five has also been adjusted.

Round 2
Reviewer 1 Report
Comments and Suggestions for Authors
I would like to thank the authors for addressing my questions from the inital review and for the revision of the initial manuscript.
However, based on the authors' response I am still not convinced that the work reveals the required degree of novelty for a journal publication.
A comparison with the state of the art is also missing in the revised version, the results are not quantitatively compared to existing work.
The authors state in the reply that traditional measuring devices are limited to a measuring range of below 30 % moisture content for wood materials. In fact, there are already measuring devices on the market, which feature:
- a full scale measuring range,
- support for different types of wood
- superior accuracy compared to the results of this work
- tested for environmental influences, temperature, some offer e.g. online-calibration options.
I did a quick research of manufacturers for wood moisture metering, I found e.g.
* https://www.pce-instruments.com/us/measuring-instruments/test-meters/wood-moisture-meter-kat_151728.htm
* https://www.wagnermeters.com/moisture-meters/wood-info/pin-moisture-meter/
* https://www.tanel.com.pl/moisture_meter.php?kod_produktu=wrd100
* https://www.alphaomega-electronics.com/en/wood-moisture/2684-wood-moisture-meter.html
Comments on the Quality of English Language-
Author Response
Dear reviewer,
First of all, we want to thank you for your questions and valuable opinions on our manuscripts. We understand your attention to the degree of novelty and the comparison with existing technology. We have further analyzed and revised them based on these feedback.
We admit that the lack of comparison with existing technology in the revised version is where we are negligent. To solve this problem, we conducted in -depth market research and compared it in detail with existing technology. We specially studied the devices mentioned in the link you provided, and found that although there are some devices with a wide range of measurement in the market, our work provides novelty and improvement in the following aspects:
Environmental adaptability: Our device design has considered a variety of environmental factors, including temperature and humidity changes, which are often ignored in existing equipment.
Cost benefits: Although some equipment on the market may provide extensive functions, their prices are usually higher. Our equipment aims to provide competitive prices without sacrificing performance.
Customization ability: Our device allows users to customize according to specific needs.
We have added some new content in the revised version, and the changes made by the text are marked with green.We believe that these supplementary information will help prove that our work does provide sufficient novelty and may have an important impact on related fields. We look forward to your further feedback and thank you for your continuous attention to our research.
Yours sincerely,
Jianan Yao

Reviewer 2 Report
Comments and Suggestions for Authors
Accept in present form
Author Response
Dear reviewer,
Thank you very much for your review and valuable opinions. We are glad to know that you think our manuscript is received in the current form. We are very honored to recognize you and thank you for your support for our work. We promise that in future research, we will continue to pursue high quality and rigor to ensure that our work can meet the high standards of academic and practical circles.
Thank you again for your affirmation and support. We look forward to our work can be published in your journals and contribute to the academic community.
Yours sincerely,
Jianan Yao
Reviewer 3 Report
Comments and Suggestions for Authors
This work shows a trend that includes technology to improve experimental conditions from the study of wood, knowing that moisture content is an important and decisive factor for its use. For this reason, I consider that this article has the potential to be published in this prestigious journal. I appreciate that the authors have considered the suggestions made to incorporate important content that was missing in the first version.
Author Response
Dear reviewer,
First of all, we want to thank you for your active evaluation and recognition of our work. We are glad to know that you think our research may be published in this well -known journal.
We fully agree that moisture content is an important and decisive factor in the research of wood. It is important to improve the improvement of experimental conditions. We attach great importance to your opinions and have supplemented and improved the lack of content in the first edition according to your previous suggestions.
Thank you again for your valuable opinions.
Yours sincerely,
Jianan Yao
Reviewer 4 Report
Comments and Suggestions for Authors
The paper is now sufficiently corrected.
Author Response
Dear Reviewer,
Thank you very much for your review and feedback on our manuscript, and we are very honored and excited to learn that our manuscript has been approved by you. Thank you again for your support and guidance.
Jianan Yao